# Uncertainty in Gradient Boosting via Ensembles

**Andrey Malinin**[*]
Yandex; HSE University
Moscow, Russia
am969@yandex-team.ru

**Liudmila Prokhorenkova**[*]
Yandex; HSE University;
Moscow Institute of Physics and Technology
Moscow, Russia
ostroumova-la@yandex-team.ru

**Aleksei Ustimenko**[*]
Yandex
Moscow, Russia
austimenko@yandex-team.ru

## Abstract

For many practical, high-risk applications, it is essential to quantify uncertainty in a model's predictions to avoid costly mistakes. While predictive uncertainty is widely studied for neural networks, the topic seems to be under-explored for models based on gradient boosting. However, gradient boosting often achieves state-of-the-art results on tabular data. This work examines a probabilistic ensemble-based framework for deriving uncertainty estimates in the predictions of gradient boosting classification and regression models. We conducted experiments on a range of synthetic and real datasets and investigated the applicability of ensemble approaches to gradient boosting models that are themselves ensembles of decision trees. Our analysis shows that ensembles of gradient boosting models successfully detect anomalous inputs while having limited ability to improve the predicted total uncertainty. Importantly, we also propose a concept of a *virtual* ensemble to get the benefits of an ensemble via only *one* gradient boosting model, which significantly reduces complexity.

## 1 Introduction

Gradient boosting (Friedman, 2001) is a widely used machine learning algorithm that achieves state-of-the-art results on tasks containing heterogeneous features, complex dependencies, and noisy data: web search, recommendation systems, weather forecasting, and many others (Burges, 2010; Caruana & Niculescu-Mizil, 2006; Richardson et al., 2007; Roe et al., 2005; Wu et al., 2010; Zhang & Haghani, 2015). Gradient boosting based on decision trees (GBDT) underlies such well-known libraries like XGBoost, LightGBM, and CatBoost. In this paper, we investigate the estimation of predictive uncertainty in GBDT models. Uncertainty estimation is crucial for avoiding costly mistakes in high-risk applications, such as autonomous driving, medical diagnostics, and financial forecasting. For example, in self-driving cars, it is necessary to know when the AI-pilot is confident in its ability to drive and when it is not to avoid a fatal collision. In financial forecasting and medical diagnostics, mistakes on the part of an AI forecasting or diagnostic system could either lead to large financial or reputational loss or to the loss of life. Crucially, both financial and medical data are often represented in heterogeneous tabular form — data on which GBDTs are typically applied, highlighting the relevance of our work on obtaining uncertainty estimates for GBDT models.

Approximate Bayesian approaches for uncertainty estimation have been extensively studied for neural network models (Gal, 2016; Malinin, 2019). Bayesian methods for tree-based models (Chipman et al., 2010; Linero, 2017) have also been widely studied in the literature. However, this research did not explicitly focus on studying *uncertainty estimation* and its applications. Some related work was

---

[*]All authors contributed equally and are listed in alphabetical order.

done by Coulston et al. (2016); Shaker & Hüllermeier (2020), who examined quantifying predictive uncertainty for random forests. However, the area has been otherwise relatively under-explored, especially for GBDT models that are widely used in practice and known to outperform other approaches based on tree ensembles.

While for classification problems GDBT models already return a distribution over class labels, for regression tasks they typically yield only point predictions. Recently, this problem was addressed in the NGBoost algorithm (Duan et al., 2020), where a GBDT model is trained to return the mean and variance of a normal distribution over the target variable $y$ for a given feature vector. However, such models only capture *data uncertainty* (Gal, 2016; Malinin, 2019), also known as *aleatoric uncertainty*, which arises due to inherent class overlap or noise in the data. However, this does not quantify uncertainty due to the model's inherent lack of knowledge about inputs from regions either far from the training data or sparsely covered by it, known as *knowledge uncertainty*, or *epistemic uncertainty* (Gal, 2016; Malinin, 2019). One class of approaches for capturing *knowledge uncertainty* are Bayesian ensemble methods, which have recently become popular for estimating predictive uncertainty in neural networks (Depeweg et al., 2017; Gal & Ghahramani, 2016; Kendall et al., 2018; Lakshminarayanan et al., 2017; Maddox et al., 2019; Smith & Gal, 2018). A key feature of ensemble approaches is that they allow overall uncertainty to be decomposed into *data uncertainty* and *knowledge uncertainty* within an interpretable probabilistic framework (Depeweg et al., 2017; Gal, 2016; Malinin, 2019). Ensembles are also known to yield improvements in predictive performance.

This work examines ensemble-based uncertainty-estimation for GBDT models. The contributions are as follows. First, we consider generating ensembles using both classical Stochastic Gradient Boosting (SGB) as well as the recently proposed Stochastic Gradient Langevin Boosting (SGLB) (Ustimenko & Prokhorenkova, 2020). Importantly, SGLB allows us to guarantee that the models are asymptotically sampled from a true Bayesian posterior. Second, we show that using SGLB we can construct a *virtual* ensemble using only *one* gradient boosting model, significantly reducing the computational complexity. Third, to understand the attributes of using ensembles-based uncertainty estimation in GBDT models, we conduct extensive analysis on several synthetic datasets. Finally, we evaluate the proposed approach on a range of real regression and classification datasets. Our results show that this approach successfully enables the detection of anomalous out-of-domain inputs. Importantly, our solution is easy to combine with any implementation of GBDT. Our methods have been implemented within the open-source CatBoost library. The code of our experiments is publicly available at `https://github.com/yandex-research/GBDT-uncertainty`.

## 2 PRELIMINARIES

**Uncertainty Estimation via Bayesian Ensembles**   In this work we consider uncertainty estimation within the standard Bayesian ensemble-based framework (Gal, 2016; Malinin, 2019). Here, model parameters $\boldsymbol{\theta}$ are considered random variables and a prior $p(\boldsymbol{\theta})$ is placed over them to compute a posterior $p(\boldsymbol{\theta}|\mathcal{D})$ via Bayes' rule:

$$p(\boldsymbol{\theta}|\mathcal{D}) = \frac{p(\mathcal{D}|\boldsymbol{\theta})p(\boldsymbol{\theta})}{p(\mathcal{D})} . \tag{1}$$

where $\mathcal{D} = \{\boldsymbol{x}^{(i)}, y^{(i)}\}_{i=1}^{N}$ is the training dataset. Each set of parameters can be considered a hypothesis or explanation about how the world works. Samples from the posterior should yield explanations consistent with the observations of the world contained within the training data $\mathcal{D}$. However, on data far from $\mathcal{D}$ each set of parameters can yield different predictions. Therefore, estimates of *knowledge uncertainty* can be obtained by examining the diversity of predictions.

Consider an ensemble of probabilistic models $\{P(y|\boldsymbol{x}; \boldsymbol{\theta}^{(m)})\}_{m=1}^{M}$ sampled from the posterior $p(\boldsymbol{\theta}|\mathcal{D})$. Each model $P(y|\boldsymbol{x}, \boldsymbol{\theta}^{(m)})$ yields a *different* estimate of *data uncertainty*, represented by the entropy of its predictive distribution (Malinin, 2019). Uncertainty in predictions due to *knowledge uncertainty* is expressed as the level of spread, or "disagreement", of models in the ensemble (Malinin, 2019). Note that exact Bayesian inference is often intractable, and it is common to consider either an explicit or implicit approximation $q(\boldsymbol{\theta})$ to the true posterior $p(\boldsymbol{\theta}|\mathcal{D})$. While a range of approximations has been explored for neural network models (Gal & Ghahramani, 2016; Lakshminarayanan et al., 2017; Maddox et al., 2019)[1], to the best of our knowledge, limited work

---

[1] A full overview is available in (Ashukha et al., 2020; Ovadia et al., 2019).

has explored Bayesian inference for gradient-boosted trees. Given $p(\boldsymbol{\theta}|\mathcal{D})$, the *predictive posterior* of the ensemble is obtained by taking the expectation with respect to the models in the ensemble:

$$P(y|\boldsymbol{x}, \mathcal{D}) = \mathbb{E}_{p(\boldsymbol{\theta}|\mathcal{D})}\big[P(y|\boldsymbol{x}; \boldsymbol{\theta})\big] \approx \frac{1}{M}\sum_{m=1}^{M} P(y|\boldsymbol{x}; \boldsymbol{\theta}^{(m)}), \ \boldsymbol{\theta}^{(m)} \sim p(\boldsymbol{\theta}|\mathcal{D}). \tag{2}$$

The entropy of the predictive posterior estimates *total uncertainty* in predictions:

$$\mathcal{H}\big[P(y|\boldsymbol{x}, \mathcal{D})\big] = \mathbb{E}_{P(y|\boldsymbol{x}, \mathcal{D})}\big[-\ln P(y|\boldsymbol{x}, \mathcal{D})\big]. \tag{3}$$

*Total uncertainty* is due to both *data uncertainty* and *knowledge uncertainty*. However, in applications like active learning (Kirsch et al., 2019) and out-of-domain detection it is desirable to estimate *knowledge uncertainty* separately. The sources of uncertainty can be decomposed by considering the *mutual information* between the parameters $\boldsymbol{\theta}$ and the prediction $y$ (Depeweg et al., 2017):

$$\underbrace{\mathcal{I}\big[y, \boldsymbol{\theta}|\boldsymbol{x}, \mathcal{D}\big]}_{\text{Knowledge Uncertainty}} = \underbrace{\mathcal{H}\big[P(y|\boldsymbol{x}, \mathcal{D})\big]}_{\text{Total Uncertainty}} - \underbrace{\mathbb{E}_{p(\boldsymbol{\theta}|\mathcal{D})}\big[\mathcal{H}[P(y|\boldsymbol{x}; \boldsymbol{\theta})]\big]}_{\text{Expected Data Uncertainty}}$$

$$\approx \mathcal{H}\big[\frac{1}{M}\sum_{m=1}^{M} P(y|\boldsymbol{x}; \boldsymbol{\theta}^{(m)})\big] - \frac{1}{M}\sum_{m=1}^{M} \mathcal{H}[P(y|\boldsymbol{x}; \boldsymbol{\theta}^{(m)})]. \tag{4}$$

This is expressed as the difference between the entropy of the predictive posterior, a measure of *total uncertainty*, and the expected entropy of each model in the ensemble, a measure of *expected data uncertainty*. Their difference is a measure of ensemble diversity and estimates *knowledge uncertainty*.

Unfortunately, when considering ensembles of probabilistic *regression* models $\{p(y|\boldsymbol{x}; \boldsymbol{\theta}^{(m)})\}_{m=1}^{M}$ over continuous-valued target $y \in \mathbb{R}$, it is no longer possible to obtain tractable estimates of the (differential) entropy of the predictive posterior, and, by extension, mutual information. In this cases uncertainty estimates can instead derived via the law of total variation:

$$\underbrace{\mathbb{V}_{p(y|\boldsymbol{x}, \mathcal{D})}[y]}_{\text{Total Uncertainty}} = \underbrace{\mathbb{V}_{p(\boldsymbol{\theta}|\mathcal{D})}\big[\mathbb{E}_{p(y|\boldsymbol{x}, \boldsymbol{\theta})}[y]\big]}_{\text{Knowledge Uncertainty}} + \underbrace{\mathbb{E}_{p(\boldsymbol{\theta}|\mathcal{D})}\big[\mathbb{V}_{p(y|\boldsymbol{x}, \boldsymbol{\theta})}[y]\big]}_{\text{Expected Data Uncertainty}}. \tag{5}$$

This is conceptually similar to the decomposition (4) obtained via mutual information. For an ensemble of probabilistic regression models which parameterize the normal distribution, and where each models yields a mean and standard-deviation, the total variance can be computed as follows:

$$\underbrace{\mathbb{V}_{p(y|\boldsymbol{x}, \mathcal{D})}[y]}_{\text{Total Uncertainty}} \approx \underbrace{\frac{1}{M}\sum_{m=1}^{M}\Big[\Big(\sum_{m=1}^{M}\frac{\mu_m}{M}\Big) - \mu_m\Big]^2}_{\text{Knowledge Uncertainty}} + \underbrace{\frac{1}{M}\sum_{m=1}^{M}\sigma_m^2}_{\text{Expected Data Uncertainty}} \ , \ \{\mu_m, \sigma_m\} = f(\boldsymbol{x}; \boldsymbol{\theta}^{(m)}). \tag{6}$$

However, while these measures are tractable, they are based on only first and second moments, and may therefore miss high-order details in the uncertainty. They are also not scale-invariant, which can cause issues is the scale of prediction on in-domain and out-of-domain data is very different.

**Gradient boosting** is a powerful machine learning technique especially useful on tasks containing heterogeneous features. It iteratively combines weak models, such as decision trees, to obtain more accurate predictions. Formally, given a dataset $\mathcal{D}$ and a loss function $L : \mathbb{R}^2 \to \mathbb{R}$, the gradient boosting algorithm (Friedman, 2001) iteratively constructs a model $F : X \to \mathbb{R}$ to minimize the empirical risk $\mathcal{L}(F|\mathcal{D}) = \mathbb{E}_{\mathcal{D}}[L(F(\boldsymbol{x}), y)]$. At each iteration $t$ the model is updated as:

$$F^{(t)}(\boldsymbol{x}) = F^{(t-1)}(\boldsymbol{x}) + \epsilon h^{(t)}(\boldsymbol{x}), \tag{7}$$

where $F^{(t-1)}$ is a model constructed at the previous iteration, $h^{(t)}(\boldsymbol{x}) \in \mathcal{H}$ is a weak learner chosen from some family of functionds $\mathcal{H}$, and $\epsilon$ is learning rate. The weak learner $h^{(t)}$ is usually chosen to approximate the negative gradient $-g^{(t)}(\boldsymbol{x}, y) := -\frac{\partial L(y, s)}{\partial s}\big|_{s=F^{(t-1)}(\boldsymbol{x})}$:

$$h^{(t)} = \underset{h \in \mathcal{H}}{\arg\min} \, \mathbb{E}_{\mathcal{D}}\big[\big(-g^{(t)}(\boldsymbol{x}, y) - h(\boldsymbol{x})\big)^2\big]. \tag{8}$$

A weak learner $h^{(t)}$ is associated with parameters $\phi^{(t)} \in \mathbb{R}^d$. We write $h^{(t)}(\boldsymbol{x}, \phi^{(t)})$ to reflect this dependence. The set of weak learners $\mathcal{H}$ often consists of shallow decision trees, which are models that recursively partition the feature space into disjoint regions called leaves. Each leaf $R_j$ of the tree is assigned to a value, which is the estimated response $y$ in the corresponding region. We can write $h(\boldsymbol{x}, \phi^{(t)}) = \sum_{j=1}^{d} \phi_j^{(t)} \mathbf{1}_{\{\boldsymbol{x} \in R_j\}}$, so the decision tree is a linear function of $\phi^{(t)}$. The final GBDT model $F$ is a sum of decision trees (7) and the parameters of the full model are denoted by $\boldsymbol{\theta}$.

For classification tasks, a model yields estimates *data uncertainty* if it is trained via negative log-likelihood and provides a distribution over class labels. However, classic GBDT regression models yield point predictions, and there has been little research devoted to estimating predictive uncertainty. Recently, this issue was addressed by Duan et al. (2020) via an algorithm called NGBoost (Natural Gradient Boosting), which allows estimating *data uncertainty*. NGBoost simultaneously estimates the parameters of a conditional distribution $p(y|\boldsymbol{x}, \boldsymbol{\theta})$ over the target $y$ given the features $\boldsymbol{x}$, by optimizing a proper scoring rule. Typically, a normal distribution over $y$ is assumed and negative log-likelihood is taken as a scoring rule. Formally, given an input $\boldsymbol{x}$, the model $F$ predicts two parameters of normal distribution - the mean $\mu$ and the logarithm of the standard deviation $\log \sigma$. The loss function is the expected negative log-likelihood:[2]

$$p(y|\boldsymbol{x}, \boldsymbol{\theta}^{(t)}) = \mathcal{N}(y|\mu^{(t)}, \sigma^{(t)}), \quad \{\mu^{(t)}, \log \sigma^{(t)}\} = F^{(t)}(\boldsymbol{x}). \tag{9}$$

$$\mathcal{L}(\boldsymbol{\theta}|\mathcal{D}) = \mathbb{E}_{\mathcal{D}}[-\log p(y|\boldsymbol{x}, \boldsymbol{\theta})] = -\frac{1}{N} \sum_{i=1}^{N} \log p(y^{(i)}|\boldsymbol{x}^{(i)}, \boldsymbol{\theta}). \tag{10}$$

Note that $\boldsymbol{\theta}$ denotes the concatenation of two parameter vectors used to predict $\mu$ and $\log \sigma$.

## 3 GENERATING ENSEMBLES OF GDBT MODELS

As discussed in Section 2, *knowledge uncertainty* can be estimated by considering an ensemble of models $\{p(y|\boldsymbol{x}; \boldsymbol{\theta}^{(m)})\}_{m=1}^{M}$ sampled from the posterior $p(\boldsymbol{\theta}|\mathcal{D})$. The level of diversity or "disagreement" between the models is an estimate of *knowledge uncertainty*. In this section, we consider three approaches to generating an ensemble of GBDT models. We emphasize that this section discusses *ensembles of GBDT models*, where a *each* GBDT model is itself an *ensemble of trees*.

**SGB ensembles** One way to generate an ensemble is to consider several independent models generated via Stochastic Gradient Boosting (SGB). Stochasticity is added to GBDT models via random subsampling of the data at every iteration (Friedman, 2002). Specifically, at each iteration of (8) we select a subset of training objects $\mathcal{D}'$ (via bootstrap or uniformly without replacement), which is smaller than the original training dataset $\mathcal{D}$, and use $\mathcal{D}'$ to fit the next tree instead of $\mathcal{D}$. The fraction of chosen objects is called *sample rate*. This implicitly injects noise into the learning process, effectively inducing a distribution $q(\boldsymbol{\theta})$ over such models. Thus, an *SGB ensemble* is an ensemble of independent models $\{\boldsymbol{\theta}^{(m)}\}_{m=1}^{M}$ built according to SGB with different random seeds for subsampling data. Unfortunately, there are no guarantees on how well the distribution $q(\boldsymbol{\theta})$ estimates the true posterior $p(\boldsymbol{\theta}|\mathcal{D})$.

**SGLB ensembles** Remarkably, there is a way to sample GBDT models from the true posterior $p(\boldsymbol{\theta}|\mathcal{D})$ via a recently proposed Stochastic Gradient Langevin Boosting (SGLB) algorithm (Ustimenko & Prokhorenkova, 2020). SGLB combines gradient boosting with stochastic gradient Langevin dynamics (Raginsky et al., 2017) in order to achieve convergence to the global optimum even for non-convex loss functions. The algorithm has two differences compared with SGB. First, Gaussian noise is explicitly injected into the gradients, so (8) is replaced by:

$$h^{(t)} = \underset{h \in \mathcal{H}}{\arg\min} \, \mathbb{E}_{\mathcal{D}} \left[ \left( -g^{(t)}(\boldsymbol{x}, y) - h(\boldsymbol{x}, \phi) + \nu \right)^2 \right], \nu \sim \mathcal{N}\left( 0, \frac{2}{\beta \epsilon} I_{|\mathcal{D}|} \right), \tag{11}$$

where $\beta$ is the inverse diffusion temperature and $I_{|\mathcal{D}|}$ is an identity matrix. This random noise $\nu$ helps to explore the solution space in order to find the global optimum and the diffusion temperature controls the level of exploration. Second, the update (7) is modified as:

$$F^{(t)}(\boldsymbol{x}) = (1 - \gamma \epsilon) F^{(t-1)}(\boldsymbol{x}) + \epsilon h^{(t)}(\boldsymbol{x}, \phi^{(t)}), \tag{12}$$

---

[2] Since GBDT model is determined by $\theta$, we use notation $\mathcal{L}(F|\mathcal{D})$ and $\mathcal{L}(\boldsymbol{\theta}|\mathcal{D})$ interchangeably.

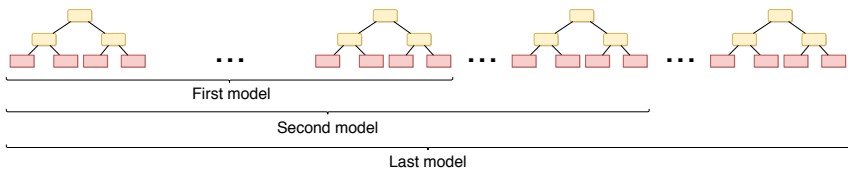

Figure 1: Virtual ensemble

where $\gamma$ is regularization parameter. If the number of all possible trees is finite (a natural assumption given that the training dataset is finite), then the SGLB parameters $\boldsymbol{\theta}^{(t)}$ at each iteration form a Markov chain that weakly converges to the stationary distribution, also called the invariant measure:

$$p_\beta^*(\boldsymbol{\theta}) \propto \exp(-\beta\mathcal{L}(\boldsymbol{\theta}|\mathcal{D}) - \beta\gamma\|\Gamma\boldsymbol{\theta}\|_2^2),\qquad(13)$$

where $\Gamma = \Gamma^T > 0$ is an implicitly defined regularization matrix which depends on a particular tree construction algorithm (Ustimenko & Prokhorenkova, 2020).

While Ustimenko & Prokhorenkova (2020) used the weak convergence to (13) to prove the global convergence, we apply this to enable sampling from the true posterior. For this purpose, we set $\beta = |\mathcal{D}|$ and $\gamma = \frac{1}{2|\mathcal{D}|}$. For the negative log-likelihood loss function (10) the invariant measure (13) can be expressed as:

$$p_\beta^*(\boldsymbol{\theta}) \propto \exp\left(\log \mathrm{p}(\mathcal{D}|\boldsymbol{\theta}) - \frac{1}{2}\|\Gamma\boldsymbol{\theta}\|_2^2\right) \propto \mathrm{p}(\mathcal{D}|\boldsymbol{\theta})\mathrm{p}(\boldsymbol{\theta}),\qquad(14)$$

which is proportional to the true posterior distribution $\mathrm{p}(\boldsymbol{\theta}|\mathcal{D})$ under Gaussian prior $\mathrm{p}(\boldsymbol{\theta}) = \mathcal{N}(0, \Gamma)$. Thus, an *SGLB ensemble* is an ensemble of independent models $\{\boldsymbol{\theta}^{(m)}\}_{m=1}^M$ generated according to the SGLB algorithm using different random seeds. In this case, asymptotically, models are sampled from the true posterior $\mathrm{p}(\boldsymbol{\theta}|\mathcal{D})$.

**Virtual SGLB ensembles** While SGB and SGLB yield ensembles of independent models, their time and space complexity is $M$ times larger than that of a single model, which is a significant overhead. Consequently, generating an ensemble requires either significantly increasing complexity or sacrificing the quality by reducing the number of training iterations. To address this, we introduce the concept of a *virtual ensemble* that enables generating an ensemble using *only one* model. This is possible since a GBDT model is itself an *ensemble of trees*. However, in contrast to random forests formed by independent trees (Shaker & Hüllermeier, 2020), the sequential nature of GBDT models implies that all trees are dependent and individual trees cannot be considered as separate models. Hence, we use "truncated" sub-models of a single GBDT model as elements of an ensemble, as illustrated in Figure 1. Notably, a virtual ensemble can be obtained using any already constructed GBDT model. Below we formally describe this procedure applied to SGLB models since in this case we can guarantee asymptotically sampling from the true posterior $\mathrm{p}(\boldsymbol{\theta}|\mathcal{D})$.

Each "truncated" model is described by the vector of parameters $\boldsymbol{\theta}^{(t)}$. As the parameters $\boldsymbol{\theta}^{(t)}$ at each iteration of the SGLB algorithm form a Markov chain that weakly convergences to the stationary distribution (14), we can consider using them as an ensemble of models. However, unlike parameters taken from different SGLB trajectories, these will have a high degree of correlation, which adversely affects the ensemble's quality. This problem can be overcome by retaining only every $K$-th set of parameters. Formally, fix $K \geq 1$ and consider a set of models $\Theta_{T,K} = \{\boldsymbol{\theta}^{(Kt)}, \left[\frac{T}{2K}\right] \leq t \leq \left[\frac{T}{K}\right]\}$, i.e., we add to $\Theta_{T,K}$ every $K$-th model obtained while constructing *one* SGLB model using $T$ iterations of gradient boosting. Choosing larger values of $K$ allows us to reduce the correlation between samples from the SGLB Markov chain. Furthermore, we do not include to the ensemble the models $\boldsymbol{\theta}^{(t)}$ with $t < T/2$ as (14) holds only asymptotically. The set of $M = \left[\frac{T}{2K}\right]$ models $\Theta_{T,K}$ is called a *virtual ensemble*. Note that virtual ensembles behave similarly to true ensembles in the limit (for large $K$ and $T$).

Importantly, we can compute the prediction of $\Theta_{T,K}$ with the same computation time as one $\boldsymbol{\theta}^{(T)}$. Indeed, when computing the prediction of one model, we have to sum up the predictions made by individual trees. To get the virtual ensemble, we only have to store the partial sums. For SGLB, we also have to account for regularization (12). Formally, according to (12), for SGLB we have

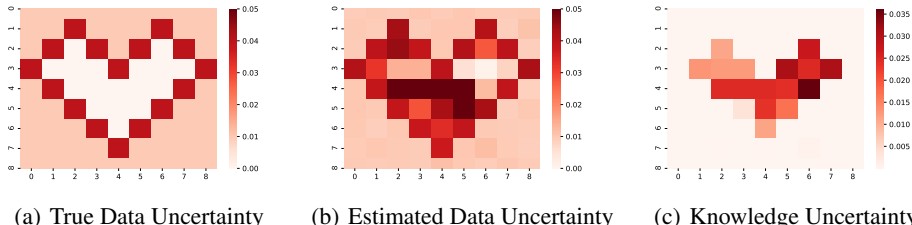

(a) True Data Uncertainty      (b) Estimated Data Uncertainty      (c) Knowledge Uncertainty

Figure 2: Uncertainty for synthetic regression dataset with two categorical features. Inside the heart (white region on the first figure) there are no training examples.

$\boldsymbol{\theta}^{(T)} = \sum_{i=1}^{T} \epsilon(1 - \gamma\epsilon)^{T-i} \boldsymbol{\phi}^{(i)}$, where $(1 - \gamma\epsilon)^{T-i}$ appears due to shrinkage. While computing $\boldsymbol{\theta}^{(T)}$ we store the partial sums $\boldsymbol{\theta}_{\leq t}^{(T)} = \sum_{i=1}^{t} \epsilon(1 - \gamma\epsilon)^{T-i} \boldsymbol{\phi}^{(i)}$. Then, any model $\boldsymbol{\theta}^{(t)}$ from $\Theta_{T,K}$ can easily be obtained from the stored values:

$$\boldsymbol{\theta}^{(t)} = \sum_{i=1}^{t} \epsilon(1 - \gamma\epsilon)^{t-i} \boldsymbol{\phi}^{(i)} = (1 - \gamma\epsilon)^{t-T} \boldsymbol{\theta}_{\leq t}^{(T)} . \tag{15}$$

## 4 ANALYSIS ON SYNTHETIC DATA

In this section, we analyze how ensemble algorithms discussed in Section 3 perform on synthetic data. The aim is to understand the attributes of ensembles of GBDT models for estimating *data* and *knowledge uncertainty* in a controllable setting.

GBDT models are usually applied to tabular data, where features are often categorical. Hence, we first generate a dataset with each example described by two categorical features $x_1, x_2$ with 9 values each, resulting in 81 possible combinations. The target depends on the features as $y = a(x_1, x_2) + \varepsilon(x_1, x_2)$, where $\varepsilon(x_1, x_2) \sim \mathcal{N}(0, b(x_1, x_2))$ and $a(x_1, x_2), b(x_1, x_2)$ are some deterministic functions. The values for $a(x_1, x_2)$ are randomly generated according to the uniform distribution over $[0, 1]$. The values for $b(x_1, x_2)$ are shown on Figure 2(a). We generate a heart-shaped dataset with this distribution: inside the heart (white region on Figure 2(a)) there are no training points, for the other cells we have 1000 examples per cell.

We train an ensemble of 10 SGLB models (each model consists of 1000 trees) and observe the following effects. First, Figure 2(b) shows the total uncertainty estimated with SGLB ensemble and we see that the models correctly capture this uncertainty in all cells containing training examples. At the same time, arbitrary values can be predicted inside the heart, as no training data constrain the models' behavior there. Second, Figure 2(c) shows that estimates of *knowledge uncertainty* allow us to detect regions that are out-of-domain and are not covered by the training data. Notably, the separation is perfect, as there is no trace of the original heart border.

To further analyze ensembles of GBDT models, we apply them to a two-dimensional classification task with continuous-valued features. We consider a 3-class spiral dataset shown on Figure 3(a); and this setting is much harder for gradient boosted trees.[3] Figure 3(b) shows the total uncertainty estimated with SGLB ensemble, while Figure 3(c) demonstrates *knowledge uncertainty*. We observe several effects. First, total uncertainty correctly detects class boundaries and 'sectors' of input space outside the training dataset. Second, looking at these 'sectors' of high uncertainty, we can better understand how GBDT ensembles work: as decision trees are *discriminative functions* (Bishop, 2006), if features have values outside the training domain, then the prediction is the same as for the "closest" elements in the dataset. In other words, the models' behavior on the boundary of the dataset is further extended to the outer regions. Third, estimates of *knowledge uncertainty* allow discrimination between out-of-domain regions and class boundaries. However, we still can see traces of the class boundaries in Figure 3(c). A possible reason is the fact that for real-valued features, near the class borders, the splitting values may vary across all models in the ensemble, resulting in non-zero estimates of *knowledge uncertainty* due to decision-boundary 'jitter'.

---

[3]To partially mitigate the difficulties, we use coordinates in rotated axes and radius as additional features.

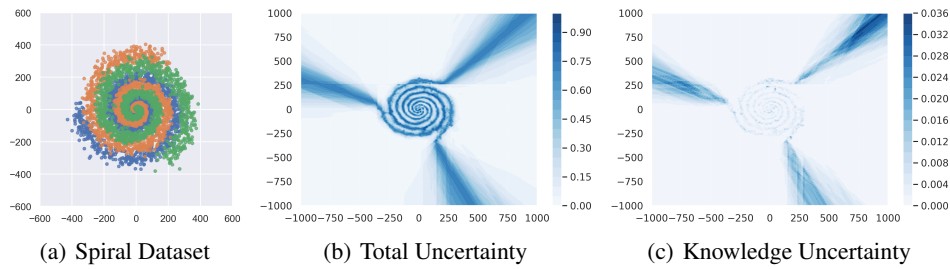

(a) Spiral Dataset  (b) Total Uncertainty  (c) Knowledge Uncertainty

Figure 3: Uncertainty for synthetic classification dataset

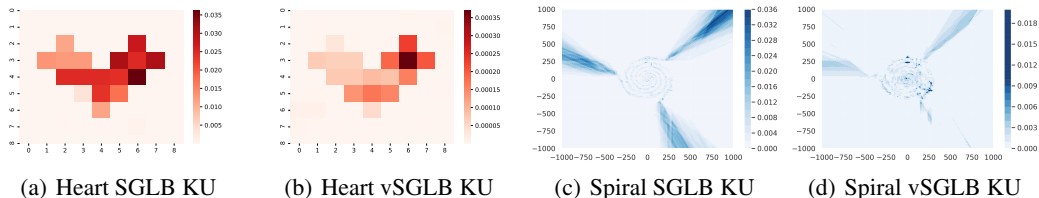

(a) Heart SGLB KU  (b) Heart vSGLB KU  (c) Spiral SGLB KU  (d) Spiral vSGLB KU

Figure 4: Comparison of SGLB and vSGLB knowledge uncertainty estimates

On both "heart" and "spiral" datasets, we observed that the absolute values of *knowledge uncertainty* are much smaller than of *data uncertainty* and therefore contribute very little to *total uncertainty*. Thus, we expect that while *knowledge uncertainty* is especially useful for detecting anomalous inputs, the proposed approaches will contribute little to error detection on top of estimates of *data uncertainty* provided by single models.

Finally, on Figure 4, we compare the performance of 'true' SGLB ensembles with the virtual SGLB ensembles (vSGBL) on both the "heart" and "spiral" datasets. The virtual ensemble is ten times cheaper to train and infer, but the ensemble members are strongly correlated. We observe that on the "heart" dataset, vSGLB perfectly detects regions not covered by training data. However, the absolute values of *knowledge uncertainty* are much smaller than for SGLB, which can be explained by the correlations. The "spiral" dataset is more challenging for both SGLB and vSGLB. While having qualitatively similar behavior, virtual ensembles struggle to detect out-of-domain regions and separate them from class boundaries. In all cases, the absolute values of *knowledge uncertainty* are far lower than for 'true' SGLB ensembles. This shows that while vSGLB yields very cheap estimates of *knowledge uncertainty* by exploiting the 'ensemble of trees' structure of GBDT models, the quality of these estimates is inferior to ensembles of independent models.

## 5 EXPERIMENTS ON CLASSIFICATION AND REGRESSION DATASETS

In this section, we evaluate the performance of ensembles of GBDT models on a range of classification and regression tasks, focusing on their ability to detect errors and out-of-domain inputs.

**Experimental setup** Our implementation of all GBDT models is based on the CatBoost library that is known to achieve state-of-the-art results in a variety of tasks (Prokhorenkova et al., 2018). Classification models yield a probability distribution over binary class labels, while regression models yield the mean and variance of the normal distribution, as discussed in Section 2. All models are trained by optimizing the negative log-likelihood.[4] We consider SGB and SGBL single models as the baselines and examine all ensemble methods defined in Section 3. Ensembles of SGB and SGLB models consist of 10 independent (with different seeds) models with 1000 trees each. The virtual ensemble vSGLB is obtained from one model with 1000 trees, where each 50th model from the interval $[501, 1000]$ is added to the ensemble. Thus, vSGLB has the same computational and

---

[4]In Appendix A.1, we compare our implementation with the original NGBoost and Deep Ensembles in terms of NLL (negative log-likelihood) and RMSE. Our implementation has comparable performance to the existing methods.

space complexity as just one SGB or SGLB model. Hyper-parameters are tuned by grid search, for details see Appendix A.2.

We compare the algorithms on several classification and regression tasks (Gal & Ghahramani, 2016; Prokhorenkova et al., 2018), the description of which is available in Appendix A.3.

While not being the focus of the current research, Random Forest (RF) models are naturally suitable for ensemble approaches. Hence, we conduct additional experiments and analyze the performance of ensemble approaches applied to RF models in Appendix C.

**Detection of errors and anomalous inputs**  We analyze whether measures of *total* and *knowledge uncertainty* can be used to detect errors and out-of-domain inputs. Error detection can be evaluated via the Prediction-Rejection Ratio (PRR) (Malinin, 2019; Malinin et al., 2020), which measures how well uncertainty estimates correlate with errors and rank-order them. The best value is 100, random is 0. Out-of-domain (OOD) detection is assessed via area under the ROC curve (AUC-ROC) (Hendrycks & Gimpel, 2016). For OOD detection, we need an OOD test-set. However, obtaining 'real' OOD examples for the datasets considered in this work is challenging, so we instead create synthetic OOD data as follows. For each dataset, we take its test set as the in-domain examples and sample an OOD dataset of the same size from the Year MSD dataset to get out-of-domain (OOD) data. The only exceptions are KDD datasets (Appetency, Churn, Upselling) and Year MSD, for which we sample OOD data from the Relative location of CT slices on axial axis Data Set (Graf et al., 2011). All numerical features in OOD data are normalized by the per-column mean and variance obtained on the in-domain training data. For categorical features, we sample a random category uniformly at random from the set of all feature's categories. *Total* and *knowledge uncertainty* are estimated via entropy of the predictive posterior (3) and mutual information (4) for classification models and via *total variance* and *variance of the mean* (5) for regression ones.

Test errors can occur due to both noise and lack of knowledge, so we expect that ranking elements by *total uncertainty* would give better values of PRR. Table 1 shows that measures of *total uncertainty* consistently yield better PPR results across all datasets. This is consistent with results obtained for ensembles of neural network models (Lakshminarayanan et al., 2017; Malinin, 2019; Malinin & Gales, 2019; Malinin et al., 2020). However, ensembles do not outperform single models. We believe this occurs for two reasons. First, due to the additive nature of boosting, GDBT models are already ensembles. Second, as we have discussed in Section 4, for GBDT models, estimates of *knowledge uncertainty* obtained via the approaches considered here contribute little to estimates of *total uncertainty*.

In contrast, Table 1 shows that measures of *knowledge uncertainty* yield superior OOD detection performance compared to *total uncertainty* in terms of AUC-ROC, which is consistent with results for non-GBDT models (Malinin, 2019; Malinin & Gales, 2019; Malinin et al., 2020).[5] The results also show that SGB and SGLB ensembles performed almost equally well. At the same time, virtual ensembling (vSGLB) performed consistently worse (with one exception) than SGB/SGLB ensembles, which is explained by the presence of strong correlations between the models in a virtual ensemble. However, in classification tasks, estimates of *knowledge uncertainty* provided by vSGLB nevertheless outperform uncertainty estimates derived from *single* SGB and SGLB models. This shows that useful measures of knowledge uncertainty *can* be derived from a *single* SGLB model by interpreting it as a virtual ensemble at *no additional computational or memory cost*. For vSGLB, the difference between classification and regression tasks can be explained by the presence or absence of categorical features. In our preliminary experiments on synthetic data, we noticed that categorical features may have a noticeable effect on the diversity of vSGLB models, and our classification datasets contain categorical features.

## 6  CONCLUSION

This work examined principled, ensemble-based uncertainty-estimation for GBDT models. Two main approaches to generating ensembles of GDBT models, where each model is itself an ensemble of trees, were considered — Stochastic Gradient Boosting (SGB) and Stochastic Gradient Langevin Boosting (SGLB). Based on SGLB, we propose constructing a *virtual* ensemble (vSGLB) by ex-

---

[5]Note that single models do not allow distinguishing between the types of uncertainty.

Table 1: Detection of errors and OOD examples for regression and classification tasks

| Dataset | | | Single | | Ensemble | | | Single | | Ensemble | | |
| --- | --- | --- | --- | --- | --- | --- | --- | --- | --- | --- | --- | --- |
| | | | SGB | SGLB | SGB | SGLB | vSGLB | SGB | SGLB | SGB | SGLB | vSGLB |
| | | | Classification % PRR (↑) | | | | | Classification % AUC-ROC (↑) | | | | |
| Adult | TU | | 72 | 72 | 72 | **72** | 72 | 53 | 50 | 52 | 51 | 51 |
| | KU | | — | — | 49 | 49 | 38 | — | — | **89** | 89 | 85 |
| Amazon | TU | | **71** | 69 | 70 | 68 | 68 | 86 | 87 | 86 | 86 | 86 |
| | KU | | — | — | 64 | 61 | 40 | — | — | **88** | 74 | 67 |
| Click | TU | | 43 | 44 | 43 | **44** | 44 | 61 | 67 | 64 | 64 | 68 |
| | KU | | — | — | 22 | 22 | 11 | — | — | 91 | **92** | 90 |
| Internet | TU | | 76 | **79** | 77 | 79 | 79 | 67 | 68 | 70 | 69 | 68 |
| | KU | | — | — | 69 | 72 | 61 | — | — | 87 | **89** | 81 |
| KDD-Appetency | TU | | 68 | 69 | **69** | 69 | 69 | 29 | 48 | 47 | 50 | 52 |
| | KU | | — | — | 64 | 54 | 14 | — | — | 90 | 91 | **93** |
| KDD-Churn | TU | | 47 | 45 | **48** | 46 | 46 | 81 | 57 | 82 | 75 | 60 |
| | KU | | — | — | 33 | 35 | 28 | — | — | **99** | 98 | 92 |
| KDD-Upselling | TU | | 56 | 56 | **57** | 57 | 56 | 53 | 51 | 62 | 60 | 47 |
| | KU | | — | — | 45 | 49 | 33 | — | — | 97 | **97** | 78 |
| Kick | TU | | 44 | **45** | 44 | 44 | 45 | 45 | 37 | 52 | 58 | 38 |
| | KU | | — | — | 34 | 34 | 20 | — | — | 98 | **98** | 89 |
| Dataset | | | Regression % PRR (↑) | | | | | Regression % AUC-ROC (↑) | | | | |
| BostonH | TU | | **45** | 45 | 44 | 45 | **46** | 70 | 68 | 71 | 69 | 64 |
| | KU | | — | — | 36 | 37 | 38 | — | — | **80** | 80 | 49 |
| Concrete | TU | | **45** | 41 | 44 | 42 | 41 | 78 | 80 | 79 | 81 | 78 |
| | KU | | — | — | 27 | 27 | 25 | — | — | **92** | 92 | 56 |
| Energy | TU | | **58** | 56 | 58 | 56 | 62 | 67 | 69 | 89 | 89 | 69 |
| | KU | | — | — | 36 | 31 | 54 | — | — | **100** | 100 | 32 |
| Kin8nm | TU | | 59 | 59 | **59** | 59 | 58 | 43 | 43 | 43 | 43 | 42 |
| | KU | | — | — | 18 | 19 | 35 | — | — | **45** | 45 | 45 |
| Naval-p | TU | | 75 | 76 | **82** | 82 | 81 | 99 | 99 | 100 | 100 | 99 |
| | KU | | — | — | 52 | 56 | 69 | — | — | **100** | 100 | 87 |
| Power-p | TU | | **30** | 32 | 31 | 33 | 32 | 48 | 47 | 51 | 49 | 47 |
| | KU | | — | — | 8 | 9 | 13 | — | — | 72 | **73** | 57 |
| Protein | TU | | 49 | 48 | **52** | 50 | 48 | 82 | 84 | 92 | 91 | 86 |
| | KU | | — | — | 30 | 29 | 12 | — | — | **99** | 99 | 94 |
| Wine-qu | TU | | **33** | 32 | 33 | 32 | 32 | 60 | 56 | 60 | 56 | 56 |
| | KU | | — | — | 25 | 19 | 9 | — | — | **74** | 72 | 49 |
| Yacht | TU | | **89** | 88 | 88 | 88 | 88 | 57 | 57 | 58 | 58 | 55 |
| | KU | | — | — | 74 | 78 | 66 | — | — | 62 | **60** | 40 |
| Year | TU | | 61 | 62 | 62 | **63** | 62 | 59 | 58 | 60 | 60 | 57 |
| | KU | | — | — | 30 | 30 | 25 | — | — | **67** | 57 | 52 |

ploiting the 'ensemble-of-trees' nature of GBDT models. Properties of the estimates of *total*, *data*, and *knowledge uncertainty* derived from these ensembles were first analyzed on synthetic data. It was shown that the proposed approach can successfully detect anomalous inputs and is especially successful on tabular data. On continuous data, detecting *knowledge uncertainty* is still possible, but it becomes harder to differentiate it with *data uncertainty* due to decision-boundary 'jitter'. Further experiments on a wide range of classification and regression datasets showed that while ensembles of GDBT models do not offer much advantage in terms of error detection, as each model is already an ensemble of trees, they do yield useful measures of *knowledge uncertainty*, which enables out-of-domain detection in both regression and classification tasks. Notably, measures of *knowledge uncertainty*, which can only be obtained via ensembles, achieve far better OOD detection performance than measures of *total uncertainty*. It is also shown that while there is little practical difference between SGB and SGLB ensembles, vSGLB performs noticeably worse. However, for classification tasks containing categorical features, vSGLB still yields useful measures of *knowledge uncertainty* at the computational time and space complexity of a *single* SGLB model. Thus, vSGLB allows us to derive the benefits of an ensemble at no additional computational and memory cost.

ACKNOWLEDGMENTS

We would like to thank Ekaterina Ermishkina and Stanislav Kirillov for implementing the proposed methods within the CatBoost library.

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

## A  EXPERIMENTAL SETUP

### A.1  OUR IMPLEMENTATION OF DATA UNCERTAINTY

As discussed in Section 2.2 of the main text, for regression we simultaneously predict the parameters $\mu$ and $\log \sigma$ of the Normal distribution. Similarly to NGBoost, we use the natural gradients. For our loss and parameterization, the natural gradient is:

$$g^{(t)}(\boldsymbol{x}, y) = \left( \mu^{(t-1)} - y, \frac{1}{2} - \frac{1}{2} \left( \frac{y - \mu^{(t-1)}}{\sigma^{(t-1)}} \right)^2 \right). \tag{16}$$

At each step of the gradient boosting procedure, we construct one tree predicting both components of $g^{(t)}$, similarly to the MultiRMSE regime of CatBoost.[6]

Recall that for classification we optimize the logistic loss.

In Table 2, we compare our implementation with NGBoots (Duan et al., 2020) and Deep Ensembles (Lakshminarayanan et al., 2017) on regression datasets. For our implementation, we consider SGB with fixed sample rate (0.5) and perform parameter tuning as described below. The best results are highlighted.

Table 2: Comparison of our implementation with existing methods

| Dataset | RMSE | | | NLL | | |
|---------|------|------|------|------|------|------|
| | Deep. Ens. | NGBoost | CatBoost | Deep. Ens. | NGBoost | CatBoost |
| Boston | $3.28 \pm 1.00$ | $\mathbf{2.94 \pm 0.53}$ | $3.06 \pm 0.68$ | $\mathbf{2.41 \pm 0.25}$ | $2.43 \pm 0.15$ | $2.47 \pm 0.20$ |
| Concrete | $6.03 \pm 0.58$ | $\mathbf{5.06 \pm 0.61}$ | $5.21 \pm 0.53$ | $3.06 \pm 0.18$ | $\mathbf{3.04 \pm 0.17}$ | $3.06 \pm 0.13$ |
| Energy | $2.09 \pm 0.29$ | $\mathbf{0.46 \pm 0.06}$ | $0.57 \pm 0.06$ | $1.38 \pm 0.22$ | $\mathbf{0.60 \pm 0.45}$ | $1.24 \pm 1.28$ |
| Kin8nm | $\mathbf{0.09 \pm 0.00}$ | $0.16 \pm 0.00$ | $0.14 \pm 0.00$ | $\mathbf{-1.20 \pm 0.02}$ | $-0.49 \pm 0.02$ | $-0.63 \pm 0.02$ |
| Naval | $\mathbf{0.00 \pm 0.00}$ | $\mathbf{0.00 \pm 0.00}$ | $\mathbf{0.00 \pm 0.00}$ | $\mathbf{-5.63 \pm 0.05}$ | $-5.34 \pm 0.04$ | $-5.39 \pm 0.04$ |
| Power | $4.11 \pm 0.17$ | $3.79 \pm 0.18$ | $\mathbf{3.55 \pm 0.27}$ | $2.79 \pm 0.04$ | $2.79 \pm 0.11$ | $\mathbf{2.72 \pm 0.12}$ |
| Protein | $4.71 \pm 0.06$ | $4.33 \pm 0.03$ | $\mathbf{3.92 \pm 0.08}$ | $2.83 \pm 0.02$ | $2.81 \pm 0.03$ | $\mathbf{2.73 \pm 0.07}$ |
| Wine | $0.64 \pm 0.04$ | $\mathbf{0.63 \pm 0.04}$ | $\mathbf{0.63 \pm 0.04}$ | $0.94 \pm 0.12$ | $\mathbf{0.91 \pm 0.06}$ | $0.93 \pm 0.08$ |
| Yacht | $1.58 \pm 0.48$ | $\mathbf{0.50 \pm 0.20}$ | $0.82 \pm 0.40$ | $1.18 \pm 0.21$ | $\mathbf{0.20 \pm 0.26}$ | $0.41 \pm 0.39$ |
| Year MSD | $\mathbf{8.89 \pm NA}$ | $8.94 \pm NA$ | $8.99 \pm NA$ | $\mathbf{3.35 \pm NA}$ | $3.43 \pm NA$ | $3.43 \pm NA$ |

### A.2  PARAMETER TUNING

For all approaches, we use grid search to tune *learning-rate* in $\{0.001, 0.01, 0.1\}$, tree *depth* in $\{3, 4, 5, 6\}$. We fix *subsample* to 0.5 for SGB and to 1 for SGLB. This is done to avoid joint randomization effects of SGB sampling and SGLB noise in gradients. We also set *diffusion-temperature* $= N$ and *model-shrink-rate* $= \frac{1}{2N}$ for SGLB.

### A.3  DATASETS

The datasets are described in Table 3. For regression, we use standard train/validation/test splits (UCI). For classification, we split the datasets into proportion 65/15/20 in train, validation, and test sets. For more details, see our GitHub repository.[7]

### A.4  STATISTICAL SIGNIFICANCE

For regression, we perform cross-validation to estimate statistical significance with paired $t$-test. In the corresponding tables, we highlight the approaches that are insignificantly different from the best one (p-value $> 0.05$).

---

[6] https://catboost.ai/docs/concepts/loss-functions-multiregression.html
[7] https://github.com/yandex-research/GBDT-uncertainty

Table 3: Datasets description

| Dataset | # Examples | # Features |
|---|---|---|
| Classification | | |
| Adult (Kohavi, 1996) | 48842 | 14 |
| Amazon (Kaggle, 2017) | 32769 | 9 |
| Click (KDD, 2012) | 399482 | 11 |
| Internet (UCI, 1997) | 10108 | 68 |
| KDD-Appetency (KDD, 2009) | 50000 | 419 |
| KDD-Churn (KDD, 2009) | 50000 | 419 |
| KDD-Upselling (KDD, 2009) | 50000 | 419 |
| Kick (Kaggle, 2011) | 72983 | 43 |
| Regression | | |
| Boston (UCI) | 506 | 13 |
| Concrete (UCI) | 1030 | 8 |
| Energy (UCI) | 768 | 8 |
| Kin8nm (UCI) | 8192 | 8 |
| Naval (UCI) | 11934 | 16 |
| Power (UCI) | 9568 | 4 |
| Protein (UCI) | 45730 | 9 |
| Wine (UCI) | 1599 | 11 |
| Yacht (UCI) | 308 | 6 |
| Year MSD (Bertin-Mahieux et al., 2011) | 515345 | 90 |

Table 4: NLL and RMSE/Error rate for regression and classification

| Dataset | Single | | Ensemble | | | Single | | Ensemble | | |
|---|---|---|---|---|---|---|---|---|---|---|
| | SGB | SGLB | SGB | SGLB | vSGLB | SGB | SGLB | SGB | SGLB | vSGLB |
| | Classification NLL (↓) | | | | | Classification % Error (↓) | | | | |
| Adult | 0.276 | 0.273 | 0.276 | **0.271** | 0.274 | **12.8** | **12.7** | 12.8 | **12.6** | **12.7** |
| Amazon | **0.141** | 0.142 | **0.140** | 0.142 | 0.143 | **4.7** | **4.6** | 4.6 | **4.5** | **4.5** |
| Click | 0.393 | 0.392 | 0.392 | **0.391** | 0.392 | 15.6 | 15.7 | 15.6 | 15.6 | 15.6 |
| Internet | 0.224 | **0.218** | 0.221 | **0.217** | 0.218 | 9.9 | 10.0 | 9.7 | 10.0 | 10.0 |
| Appetency | **0.073** | **0.073** | 0.073 | 0.073 | 0.073 | 1.8 | 1.8 | 1.8 | 1.8 | 1.8 |
| Churn | 0.235 | 0.236 | **0.233** | 0.234 | 0.235 | 7.3 | **7.2** | 7.3 | **7.2** | **7.2** |
| Upselling | **0.168** | **0.168** | **0.168** | **0.168** | **0.168** | 5.0 | 4.9 | 5.0 | 5.0 | 4.9 |
| Kick | 0.287 | 0.286 | 0.286 | **0.285** | 0.286 | 9.5 | 9.6 | 9.5 | **9.4** | 9.6 |
| Dataset | Regression NLL (↓) | | | | | Regression RMSE (↓) | | | | |
| BostonH | 2.47 | 2.52 | **2.46** | **2.50** | **2.50** | **3.06** | **3.12** | 3.04 | **3.10** | **3.27** |
| Concrete | **3.06** | 3.06 | **3.05** | **3.05** | 3.06 | 5.21 | **5.11** | 5.21 | **5.10** | 5.37 |
| Energy | **1.24** | **1.70** | 1.13 | 1.52 | **0.70** | 0.57 | **0.54** | 0.57 | **0.54** | 0.64 |
| Kin8nm | -0.63 | -0.65 | -0.63 | **-0.65** | -0.60 | 0.14 | **0.14** | 0.14 | **0.14** | 0.15 |
| Naval-p | -5.39 | -5.42 | -5.61 | **-5.65** | -5.39 | 0.00 | 0.00 | **0.00** | **0.00** | 0.00 |
| Power-p | 2.72 | 2.71 | **2.66** | **2.66** | 2.69 | 3.55 | **3.56** | 3.52 | **3.54** | 3.64 |
| Protein | 2.73 | 2.73 | **2.61** | 2.64 | 2.70 | 3.92 | **3.96** | 3.90 | **3.93** | 4.02 |
| Wine-qu | **0.93** | 0.99 | **0.92** | 0.98 | 0.96 | **0.63** | 0.65 | 0.63 | 0.65 | 0.66 |
| Yacht | **0.41** | 0.38 | **0.27** | **0.32** | 0.51 | **0.82** | 0.84 | 0.83 | 0.84 | 0.97 |
| Year | 3.43 | 3.43 | 3.41 | **3.40** | 3.42 | 8.99 | 8.96 | 8.97 | **8.94** | 8.98 |

For classification (and Year MSD), we measure statistical significance for NLL and error/RMSE on the test set. In the corresponding tables, the approaches that are insignificantly different from the best one are highlighted. For PRR and AUC-ROC (for classification and Year MSD), we highlight the best value.

## B   ADDITIONAL EXPERIMENTAL RESULTS

In Table 4, we compare ensemble approaches with single models in terms of NLL and error rate for classification and in terms of NLL and RMSE for regression tasks. Results for NLL demonstrate an advantage of ensembling approaches compared to single models. However, in some cases the difference is not significant, which can be explained by the additive nature of boosting: averaging several tree ensembles gives another (larger) tree ensemble. Thus, improved NLL can result from the increased complexity of ensemble models. We can make a similar conclusion from the results for RMSE and error rate.

## C   COMPARISON WITH RANDOM FOREST

Our paper specifically focuses on uncertainty estimation in Gradient Boosted Decision Trees (GBDT) models. However, some related work was done for quantifying uncertainty in random forests (Coulston et al., 2016; Shaker & Hüllermeier, 2020), which are also ensembles of decision trees. Thus, for completeness, we also analyze how ensemble approaches perform in combination with random forests.

In these experiments, we use the scikit-learn implementation of random forests (Pedregosa et al., 2011). We limit the maximum depth to 10 and keep all other parameters default. For categorical features, we use leave-one-out encoding.

Unlike GBDT, where trees are added to correct the previous model's mistakes, random forests (RF) consist of decision trees that are *independently* trained on bootstrapped sub-samples of the dataset. Hence, for *knowledge uncertainty* we can divide RF into several *independent* parts, each consisting of several trees. Drawing a parallel to virtual SGLB, we call this approach vRF (virtual RF) since it allows estimating *knowledge uncertainty* using only one trained random forest model. In our experiments with vRF, we divide one RF model into 10 independent parts, each consisting of 100 trees. Similarly, one can also construct an ensemble of several independently trained random forest models, which is expected to be a stronger baseline. However, we expect a small difference between vRF and an ensemble of random forests, as there are, a priori, no correlations between trees both in a single model and across multiple RF models.

In Tables 5 and 6, we compare the predictive performance of random forests (both individual models and explicit ensembles of multiple models) to SGLB individual and ensemble models on classification and regression tasks. The results show that generally GBDT models outperform random forest models in terms of classification error rate and NLL. Note that we cannot calculate NLL for RF regression models as they are not naturally probabilistic (do not yield a predicted variance). As a result, they are unable to estimate *data uncertainty*, and therefore we can only obtain estimates of *knowledge uncertainty*.

Table 7 compares SGLB and RF ensembles in terms of error detection (PRR) and out-of-domain input detection (ROC-AUC). One can see that SGLB usually outperforms RF, especially for OOD detection. Notably, as we expected, for OOD detection vRF and RF give similar results. Thus, we conclude that for random forests, a virtual ensemble is a good and cheap alternative to the true one.

Table 5: Comparison with random forest: NLL and error rate for classification

| Dataset | Single | | Ensemble | | Single | | Ensemble | |
|---|---|---|---|---|---|---|---|---|
| | SGLB | RF | SGLB | RF | SGLB | RF | SGLB | RF |
| | Classification NLL (↓) | | | | Classification % Error (↓) | | | |
| Adult | 0.273 | 0.300 | **0.271** | 0.300 | **12.7** | 13.9 | **12.6** | 13.9 |
| Amazon | **0.142** | 0.183 | **0.142** | 0.183 | **4.6** | 5.6 | **4.5** | 5.6 |
| Click | 0.392 | 0.411 | **0.391** | 0.411 | **15.7** | 16.0 | **15.6** | 16.0 |
| Internet | **0.218** | 0.275 | **0.217** | 0.274 | **10.0** | 11.2 | **10.0** | 11.0 |
| KDD-Appetency | **0.073** | 0.083 | **0.073** | 0.083 | **1.8** | 1.8 | 1.8 | 1.8 |
| KDD-Churn | 0.236 | 0.249 | **0.234** | 0.249 | **7.2** | 7.3 | **7.2** | 7.3 |
| KDD-Upselling | **0.168** | 0.202 | **0.168** | 0.202 | **4.9** | 7.4 | **5.0** | 7.4 |
| Kick | 0.286 | 0.311 | **0.285** | 0.311 | 9.6 | 10.4 | **9.4** | 10.4 |

Table 6: Comparison with random forest: RMSE for regression

| Dataset | Single | | Ensemble | |
|---|---|---|---|---|
| | SGLB | RF | SGLB | RF |
| BostonH | **3.12** | 2.98 | **3.10** | 2.98 |
| Concrete | **5.11** | 4.96 | **5.10** | 4.95 |
| Energy | 0.54 | **0.50** | 0.54 | **0.50** |
| Kin8nm | **0.14** | 0.15 | **0.14** | 0.15 |
| Naval-p | 0.00 | 0.00 | **0.00** | 0.00 |
| Power-p | **3.56** | 3.53 | **3.54** | 3.53 |
| Protein | 3.96 | 4.19 | **3.93** | 4.19 |
| Wine-qu | 0.65 | **0.58** | 0.65 | **0.58** |
| Yacht | **0.84** | 0.84 | **0.84** | 0.84 |
| Year | 8.96 | 9.43 | **8.94** | 9.43 |

Table 7: Comparison with random forest: detection of errors and OOD examples for regression and classification (virtual and true ensembles)

| Dataset | | vSGLB | vRF | SGLB | RF | vSGLB | vRF | SGLB | RF |
|---|---|---|---|---|---|---|---|---|---|
| | | Classification % PRR (↑) | | | | Classification % AUC-ROC (↑) | | | |
| Adult | TU | 72 | 70 | **72** | 70 | 51 | 58 | 51 | 58 |
| | KU | 38 | 20 | 49 | 22 | 85 | 86 | **89** | 87 |
| Amazon | TU | 68 | 63 | **68** | 64 | **86** | 48 | 86 | 49 |
| | KU | 40 | 45 | 61 | 52 | 67 | 55 | 74 | 53 |
| Click | TU | 44 | 36 | **44** | 37 | 68 | 71 | 64 | 71 |
| | KU | 11 | 20 | 22 | 21 | 90 | 81 | **92** | 80 |
| Internet | TU | 79 | 69 | **79** | 68 | 68 | 72 | 69 | 72 |
| | KU | 61 | 38 | 72 | 36 | 81 | 79 | **89** | 79 |
| KDD-Appetency | TU | 69 | 56 | **69** | 56 | 52 | 82 | 50 | 81 |
| | KU | 14 | 34 | 54 | 34 | 93 | 97 | 91 | **98** |
| KDD-Churn | TU | **46** | 39 | 46 | 39 | 60 | 75 | 75 | 78 |
| | KU | 28 | 13 | 35 | 9 | 92 | 93 | **98** | 94 |
| KDD-Upselling | TU | 56 | **66** | 57 | 66 | 47 | 73 | 60 | 72 |
| | KU | 33 | 45 | 49 | 44 | 78 | 90 | **97** | 92 |
| Kick | TU | **45** | 38 | 44 | 38 | 38 | 64 | 58 | 63 |
| | KU | 20 | 27 | 34 | 29 | 89 | 89 | **98** | 89 |
| Dataset | | Regression % PRR (↑) | | | | Regression % AUC-ROC (↑) | | | |
| BostonH | TU | **46** | — | 45 | — | 64 | — | 69 | — |
| | KU | 38 | **53** | 37 | **55** | 49 | 75 | **80** | 76 |
| Concrete | TU | **41** | — | 42 | — | 78 | — | 81 | — |
| | KU | 25 | **43** | 27 | **42** | 56 | 80 | **92** | 80 |
| Energy | TU | **62** | — | 56 | — | 69 | — | 89 | — |
| | KU | 54 | 40 | 31 | 41 | 32 | 100 | **100** | 100 |
| Kin8nm | TU | 58 | — | **59** | — | 42 | — | 43 | — |
| | KU | 35 | 33 | 19 | 33 | 45 | **48** | 45 | **48** |
| Power-p | TU | **32** | — | 33 | — | 47 | — | 49 | — |
| | KU | 13 | 21 | 9 | 22 | 57 | 66 | **73** | 65 |
| Protein | TU | 48 | — | **50** | — | 86 | — | 91 | — |
| | KU | 12 | 40 | 29 | 40 | 94 | 91 | **99** | 92 |
| Wine-qu | TU | **32** | — | 32 | — | 56 | — | 56 | — |
| | KU | 9 | **35** | 19 | **32** | 49 | 74 | 72 | 74 |
| Yacht | TU | **88** | — | 88 | — | 55 | — | 58 | — |
| | KU | 66 | 79 | 78 | 81 | 40 | 52 | **60** | 52 |
| Year | TU | 62 | — | **63** | — | 57 | — | 60 | — |
| | KU | 25 | 28 | 30 | 27 | 52 | 74 | 57 | **74** |

