# OpenReview forum: "Uncertainty in Gradient Boosting via Ensembles"
_ICLR.cc/2021/Conference — ICLR 2021 Poster_

### Official Review · AnonReviewer3 · 2020-10-28
**Measuring knowledge uncertainty with GBDT**

**Rating:** 6
**Confidence:** 3

**Review:**

Summary:
The authors propose to apply the idea of Bayesian ensemble methods to (tree-based) gradient boosting methods so as to be able to measure the knowledge uncertainty (e.g., to detect anomaly or out-of-domains samples) while typically only data uncertainty (e.g. related to noise in the data) is considered.

Overal comment:
The paper is well-written and the amount of experiments is impressive. However, the proposed approach raises several concerns addressed below and regarding the novelty mainly (comment (b)) and the actual performances (comment (d)). I think the paper could be accepted if comments (b) and (d) are addressed.

Major comments:
(a) The motivation is well explained and I agree that not knowing when a machine learnt predictor makes a prediction with certainty or not is the key of the trust in such models.
(b) It is unclear how the approach for estimating uncertainty is novel with respect to what was made with other ensemble approaches as Section II refers to preliminary works but Section III (except virtual ensembles) refers to ensemble of GB models and I am not sure it is an actual contribution of this paper. Please clarify what's new and what already exists in Section III and how much your approach differs / improves / modifies what has already been suggested by the uncertainties estimation via Bayesian Ensembles.
(c) As suggested by Figure 1, it appears that a virtual ensemble is built recursively by adding the one tree model to the previous ensemble and making hence a new object of the virtual ensemble. This suggests that models are not independent and intrinsically give a much higher weight on the first model for instance. The motivation behind the building of the virtual ensemble (the choice of T/2 in the text for instance) is not very clear either. Please clarify (a) if Figure 1 is not overly simplistic and (b) the motivation of the making of a virtual ensemble (especially the value T/2).
(d) It would have been interesting  (and more convincing) to have results (of Table 1) for other non-gradient boosting techniques to convince the soundness of the proposed approach and how it compares with other techniques.

Minor comments:
- typo : "it's (applications)" in the first section.
- Typically, loss functions are defined as function $R^2 \rightarrow R^+$ justifying the "negative" log-likelihood. Please check.

---

> ### Author Response · Authors · 2020-11-13
> **Response to Reviewer 3**
>
> Thank you very much for the positive feedback! Let us reply to the major comments:
>
> (b) What we feel is novel and consider our core contribution is the investigation of Bayesian (ensemble-based) uncertainty estimation for GBDT models and the various attributes that this interaction entails. There has been no prior work, to our knowledge, on this topic. By itself, ensemble-based uncertainty estimation has been explored in detail in Deep Learning. Similarly, GBDT models have also been explored.
>
> With regards to Section III, we specifically examine ways in which we can construct ensembles of GBDT models such that the models have a low degree of correlation (SGB/SGLB). Importantly, we show that SGLB models have theoretical guarantees and are very suitable for the Bayesian approach. We also make use of the ensemble-like structure of a single GBDT model to explore virtual SGLB ensembles, which are computationally cheaper but perform worse due to significant levels of self-correlation. The introduction and analysis of virtual ensembles is also a contribution of our work.
>
> (c) A virtual ensemble is a way of getting an ensemble of several models given only one trained model. In other words, it is assumed that we do not have computational and memory resources to train several independent models. Hence, the ensemble elements will be dependent.
> Due to the sequential construction of GBDT models, the first trees are more important and their error is only compensated by further trees. Hence, to get a reasonable approximation of the target dependency, we can only consider prefixes (sums of the first several trees) as models in our ensemble. This is the reason why several first trees are present in all models.
> The first model in our ensemble consists of T/2 trees. This is done in order to get a reasonably strong model. For instance, if we assume that a model consists of only the first tree, that would be a very rough approximation. Then, each next model is obtained from the previous one by adding K more trees. Larger values of K allow us to reduce the dependency between models. For instance, if we add only one tree, then the models are extremely dependent. In practice, we usually construct 10 models, so we add T/20 trees at each step.
> Finally, let us note that vSGLB is built on top of SGLB. According to eq. (11), at each step of SGLB construction, the current model is being shrunk since it is multiplied by a value that is less than 1. This allows us to reduce the importance of first trees in the final model. It also helps to get models that are less correlated - new trees will compensate for this shrinkage.
> We will add more explanations about virtual ensembles to the text.
>
> (d) We are following the suggestion of Reviewer 4 and setting up a RandomForest baseline.

---

> > ### Author Response · Authors · 2020-11-22
> > **We updated the paper**
> >
> > We’ve just updated the paper.
> >
> > - We added more details about virtual ensembles to Section 3.
> > - We also conducted additional experiments comparing uncertainty estimates for GBDT with Random Forests; see Appendix D for the details.
> > - Regarding the comment on loss functions, let us note that not all loss functions are strictly positive. In our case, the NLL in the regression case can be less than 0, as we are minimizing the negative logarithm of the density. If the density is larger than 1, then its negative logarithm is less than 0.
> >
> > If there are any more comments or questions, we will be happy to address them!

---

### Official Review · AnonReviewer1 · 2020-10-29
**UNCERTAINTY INGRADIENTBOOSTINGVIAENSEMBLES**

**Rating:** 6
**Confidence:** 4

**Review:**

This work examines a probabilistic ensemble-based framework for deriving uncertainty estimates in the predictions of gradient boosting classification and regression models. As the authors have said, predictive uncertainty is sometimes a must have feature for high-risk application of machine learning techniques.

The authors conducted a range of experiments on both synthetic and real datasets and investigated the applicability of ensemble approaches to gradient boosting models that are themselves ensembles of decision trees. The results seem to suggest these ensembles were able to detect anomaly input. The authors have also introduced the idea/concept of virtual ensemble by exploiting the ‘ensemble-of-trees’ nature of GBDT models. , which is interesting too.

In general I think this paper is trying to tackle a significant problem, so it would be good if the authors would give some real world examples of possible applications when uncertainty estimates in the predictions is essential. Another aspect is, as the authors have mentioned, boosting trees itself is an ensemble, I would expect the authors explain a bit more on why we need ensemble of ensembles rather than working out uncertainty estimates just using the trees in a single boosting-tree ensemble?

---

> ### Author Response · Authors · 2020-11-13
> **Response to Reviewer 1**
>
> Thank you very much for the positive feedback! Regarding the comments:
> 1. Uncertainty estimation is particularly important in high-risk applications of machine learning. For example, in self-driving cars - it is necessary to know when the AI-pilot is confident in its ability to drive and when it isn’t. Other high-risk applications are financial forecasting and medical diagnostics. In these cases, mistakes on the part of an AI forecasting or diagnostic system could either lead to large financial or reputational loss or to the loss of life. Crucially, both financial and medical data is often represented in heterogeneous tabular form - data on which GBDTs are typically applied, which highlights the significance of our work on obtaining uncertainty estimates for GBDT models.
> 2. A single GBDT model is an ensemble of individual trees, but each tree only corrects errors of the previously built model (in contrast to, e.g., random forest) and, therefore, cannot be considered as a separate model by itself. However, there is a way to create an ensemble using only one model, and this is exactly what vSGLB does. Note that vSGLB models are correlated and may therefore yield worse estimates of knowledge uncertainty compared to independent GBDT models (ensemble of ensembles). Hence, we also consider true ensembles of SGB or SGLB to get uncorrelated models, which yields superior uncertainty estimation performance at the cost of extra memory and training/inference time.

---

> > ### Author Response · Authors · 2020-11-22
> > **We updated the paper**
> >
> > We’ve just updated the paper.
> >
> > In particular, we extended the introduction by adding more motivation and practical examples.  We also conducted additional experiments comparing the proposed approach for GBDT with Random Forests, as requested by Reviewer 4.
> >
> > If there are any more comments or questions, we will be happy to address them!

---

### Official Review · AnonReviewer4 · 2020-10-29
**Good paper on a topic that needs further exploration**

**Rating:** 7
**Confidence:** 4

**Review:**

Summary:
The paper explores a strategy of building ensembles of Gradient Boosting Decision Trees (GBDTs) to improve capturing total uncertainty and knowledge uncertainty to better detect errors and out of domain data points. Overall, I vote for accepting. I think while the results are not super impressive, they are encouraging, and I would like to see more research exploring this area.

Pros:
1. I think error detection and OOD detection are an under explored area in the context of tabular data. The problem itself is important in terms of practicality.
2. The main idea of the paper is to build ensembles of GBDTs with two flavors of gradient descent. The intuition that ensembles should improve uncertainty estimation is natural, and something worth exploring.
3. The main improvement in terms of results is the improved out-of-domain data detection with SGLBs. Other than that, the results aren’t really better than a single GBDT for error detection.
4. The number of experiments done is quite comprehensive with a good mix of synthetic and real datasets.
5. It’s disheartening that even with ensembles of GBDTs, the PRR is quite low in most datasets. This is not the authors’ fault of course. I think this is an important contribution to the literature so that the community is aware it does not work for error detection.

Cons:
1. The virtual ensemble doesn’t quite give as good performances as SGLBs which the authors were emphasizing quite a bit.
2. I couldn’t find any description on how the knowledge uncertainty was calculated practically (coding wise).
3. I think it is important to at least show how Random forests or ExtraTreeForests are performing in terms of Precision-rejection ratio (PRR), and OOD detection. These ensemble methods typically perform well, so even if they perform worse here, that would be a benchmark on how much SGLBs are improving upon.

I would request the authors to address the cons during the rebuttal period.

---

> ### Author Response · Authors · 2020-11-13
> **Response to Reviewer 4**
>
> Thank you very much for the positive feedback!
>
> Regarding the comments:
> 1. Indeed, virtual ensembles are usually worse than the true ones due to the correlation between ensemble elements. However, there is a compromise between training and inference time and quality of uncertainty estimate. If computational resources are not limited, then one can use a true SGLB ensemble. If there are no resources on any computational overhead, then vSGBL is the only option, which still gives a sensible knowledge uncertainty estimate, as shown in Figure 4b.
> 2. We will add algorithmic details on how to calculate all forms of uncertainty. In short, to estimate knowledge uncertainty for regression, one needs to compute the variance of mean values predicted by models (elements of an ensemble). For classification, knowledge uncertainty is the difference between total and data uncertainties. Data uncertainty is the mean value of entropies of individual models’ predictions, while total uncertainty is the entropy of the mean prediction. This corresponds to equations (4) and (5) in the paper, but we agree that we need to write this computation explicitly.
> 3. Thank you for your suggestion! We will set up a RandomForest model as a baseline.

---

> > ### Author Response · Authors · 2020-11-22
> > **We updated the paper**
> >
> > We’ve just updated the paper.
> > - We added details about how knowledge uncertainty can be computed given an ensemble of models (see Section 2, equations (4) and (6)). Does this help, or is it better to add additional explanations (like pseudocode in addition to the formula)?
> > - We conducted additional experiments comparing SGLB with Random Forests. Since our paper is focused on GBDT approaches, we added a detailed comparison to Appendix D. There is one dataset where the true ensembles of RF models outperform other approaches for OOD detection. However, in general, SGLB ensembles (both true and virtual) have better performance.
> >
> > If there are any more comments or questions, we will be happy to address them!

---

### Official Review · AnonReviewer2 · 2020-11-02
**Well written with important practical contribution**

**Rating:** 7
**Confidence:** 4

**Review:**

This paper studied the uncertainty estimation in GBDT method. The authors described 3 methods to estimate the uncertainty. With SGB, the estimation is achieved by training multiple models using data sub-samples. With SGLB, the authors derived that we can estimate the posterior distribution of the model parameters. These two methods both have the disadvantage that the training time is multiplicative of the number of trained models. To address this issue, the authors proposed an improvement to SGLB which they call virtual SGLB. The main idea is to use a subset of trees in a GBDT as a model sample, so that we can train a single model but still able to estimate the uncertainty.

I find the paper clearly written and well organized. The theoretical formulation of the proposed methods makes sense and clear to follow. A natural concern of the vSGLB method is the correlation between the sub-trees in GBDT. Indeed, the experiments show that vSGLB is significantly worse in uncertainty estimation in real datasets. Nevertheless, I think the authors make a good case about the trade-off between a fast training time and a good uncertainty estimation.

Uncertainty estimation is an increasingly important topic in machine learning application. GBDT method is one of the most commonly used ML method in application. This paper propose a fast approach and a more accurate approach for uncertainty estimation when using GBDT. Thus I think it would be an interesting read to ML practitioners especially.

---

> ### Author Response · Authors · 2020-11-13
> **Response to Reviewer 2**
>
> Thank you very much for the positive feedback, we really appreciate it!

---

> > ### Author Response · Authors · 2020-11-22
> > **We updated the paper**
> >
> > We’ve just updated the paper. In particular, we conducted additional experiments comparing the proposed approach for GBDT with Random Forests, as requested by Reviewer 4. These results are presented in appendix D.
> >
> > If there are any more comments or questions, we will be happy to address them!

---

### Author Response · Authors · 2020-11-24
**Summary of Changes**

Summary of Changes

For clarity, we add a summary of the changes and updates we have made to the paper.

Experimental:

a. We have added a comparison to random forests. We consider ensembles of trees in a single random forest model as well as ensembles of several random forest models. The results and analysis are provided in Appendix D. The results show that in most cases (except for one dataset), SGLB ensembles (both true and virtual) have better performance. In general, GDBT models also yield superior predictive performance to RF models.

Clarity of text:

a. We have updated the introduction and discussed the applications of our approach in greater detail.

b.We added details about how knowledge uncertainty can be computed given an ensemble of models (see Section 2, equations (4) and (6)).

c. We have clarified our description of virtual SGLB ensembles.


We hope that we have satisfactorily addressed all of your concerns during this rebuttal process. If there are any more comments or questions, we will be happy to address them.

Sincerely, Authors

---

### Decision · Program_Chairs · 2021-01-07
**Final Decision**

**Decision:**

Accept (Poster)

**Comment:**

The authors design a framework to estimate the uncertainties in the predictions of gradient boosting models, for both classification and regression. The framework contains several methods, some that use sub-sampling on data to calculate the estimation, and some that use sub-sampling on the trees within one single gradient boosting model (i.e. virtual ensemble) to calculate the estimation. The different methods reveal the trade-off between faster calculation and good uncertainty estimation. The authors conduct extensive empirical study to demonstrate the validity of the designed framework.

The reviewers agree that the paper is well-written on a very important topic of machine learning in practice. The authors have done a great job addressing the comments from the reviewers, including the comparison to random forest, and adding more motivating examples. The reviewers believe that the work marks a good starting point for addressing this important topic. Nevertheless, the reviewers have some concerns that the results are promising but not impressive yet, and the performance of the virtual ensemble is a bit discouraging.